# Acute Responses of Novel Cardiac Biomarkers to a 24-h Ultra-Marathon

**DOI:** 10.3390/jcm8010057

**Published:** 2019-01-08

**Authors:** Aleksandra Żebrowska, Zbigniew Waśkiewicz, Pantelis T. Nikolaidis, Rafał Mikołajczyk, Damian Kawecki, Thomas Rosemann, Beat Knechtle

**Affiliations:** 1Department of Physiological and Medical Sciences, Academy of Physical Education, Mikołowska Street 72a, 40-065 Katowice, Poland; a.zebrowska@awf.katowice.pl (A.Ż.); r.mikolajczyk@awf.katowice.pl (R.M.); 2Department of Team Sports Games, Academy of Physical Education in Katowice, Mikołowska Street 72a, 40-065 Katowice, Poland; z.waskiewicz@awf.katowice.pl; 3Department of Sports Medicine and Medical Rehabilitation, Sechenov University, Moscow 119991, Russia; 4Laboratory of Exercise Testing, Hellenic Air Force Academy, Dekelia, Thermopylon 7, 18450 Nikaia, Greece; pademil@hotmail.com; 5Second Department of Cardiology, School of Medicine with the Division of Dentistry in Zabrze, Medical University of Silesia, Skłodowskiej, Curie 10 Street, 41-800 Zabrze, Poland; d.kawecki@sum.edu.pl; 6Institute of Primary Care, University of Zurich, 8091 Zurich, Switzerland; thomas.rosemann@usz.ch; 7Medbase St. Gallen Am Vadianplatz, 9001 St. Gallen, Switzerland

**Keywords:** human performance, myocardium ischemia, ultra-endurance

## Abstract

The aim of the present study was to examine the acute effect of an ultra-endurance performance on N-terminal pro-brain natriuretic peptide (NT-proBNP), cardiac specific troponin T (cTnT), creatinine kinase-myocardial band (CK-MB), high sensitive C-reactive protein (hsCRP), ischemia modified albumin (IMA), heart-type fatty acid binding protein (H-FABP) and cardiovascular function. Cardiac biomarkers were evaluated in 14 male ultra-marathoners (age 40 ± 12 years) during a 24 h ultra-marathon at five points (i.e., Pre-race; Marathon, 12-h run, 24-h run, and 48-h post-race). All subjects underwent baseline echocardiography assessment at least 10 days prior to the ultra-marathon and 48 h post-race. The average distance covered during the race was 149.4 ± 33.0 km. Running the ultra-marathon led to a progressive increase in hsCRP and H-FABP concentrations (*p* < 0.001). CK-MB and cTnT levels were higher after a 24-h run compared to pre-race (*p* < 0.05). Diastolic function was altered post-race characterized by a reduction in peak early to late diastolic filling (*p* < 0.01). Running an ultra-marathon significantly stimulates specific cardiac biomarkers; however, the dynamic of secretion of biomarkers linked to myocardium ischemia were differentially regulated during the ultra-marathon race. It is suggested that both exercise duration and intensity play a crucial role in cardiovascular adaptive mechanisms and cause risk of cardiac stress in ultra-marathoners.

## 1. Introduction

Findings from most scientific studies have suggested a positive relationship between physical activity and health benefits [1] with aerobic exercise increasing cardiorespiratory fitness [2], improving lipid profile [3] and constituting the most effective stimulus for adaptive changes in cardiovascular hemodynamic [4]. By contrast, in some cases prolonged exertion might have serious pathophysiological consequences including pulmonary hypertension [5], cardiovascular complications [6] and exercise-induced arterial hypoxemia [7]. The popularity of ultra-distance running might explain the increased number of studies on physiological and biochemical cardiac dysfunction in elite as well as recreational athletes [8,9,10,11]. Echocardiography measurements as well as cardiac biomarkers showed that participation in ultra-endurance events was associated with significant changes in cardiac function [12,13].

Considering potential negative effects on the heart, several studies documented echocardiographic abnormalities after long-distance running races including altered left and right ventricular functions and the appearance of wall motion abnormalities [14,15,16]. Similarly, a number of studies investigated changes in biomarkers associated with cardiac dysfunction as a consequence of marathon and ultra-marathon participation [17,18,19]. The increase in cardiac biomarkers as a consequence of extreme sports participation was related to echocardiography evidence of cardiac dysfunction and injury [20,21,22,23].

A transient increase in cardiac biomarkers has been documented for N-terminal pro-brain natriuretic peptide (NT-proBNP) and mid-regional pro-atrial natriuretic peptide (MRproANP), and these changes were primarily associated with myocardial pressure overload and wall stretch [18,22]. Further, several studies used conventional biomarkers such as creatine kinase myocardial band (CKMB) and cardiac specific troponin (cTnT) associated with hypoxia-induced myocardial injury [20,24,25,26]. However, the importance of the use ischemia modified albumin (IMA) and heart-type fatty acid binding protein (H-FABP) measurements as new markers to detect or exclude an early-stage myocardial ischemia has also been available [27,28,29,30].

There has been evidence that hypertrophic cardiomyopathy was a frequent cause of sudden cardiac death among endurance athletes, followed by cardiac anomalies, myocarditis, and arrythmogenic diseases [31]. Nevertheless, research results on ultra-marathon races were controversial and limited to observing the main cardiac biomarkers levels and their association with runners’ age, finish times [20], echocardiography [14], training status [21,32] or altitude variation [33]. The role of biochemical changes of the heart in response to marathon to ultra-marathon races up to 280 km [8] or 308 km [15] has been studied in literature; however, the consequences of this change in heart function remained mostly unexplored.

Participation in long-distance running has been on the rise, especially among the aging population [34,35]. Hence, a number of investigations have been conducted to determine the cardiovascular effects and risk factors of cardiac muscle injury in marathon and ultra-marathon runners [7,13,15,36,37]. Considering the fact that the risk of cardiac mortality increases among amateur long-distance runners [31], it seemed important to investigate markers that allow an early identification of increased cardiac risk. In previous studies of the assessment of myocardial dysfunction the conventional cardiac biomarkers (e.g., cTnT, CKMB, myoglobin, NTproBNP) have been analyzed [24,38,39,40]. These markers had different release times [25] and different times of reaching maximal concentrations [24]. It has been hypothesized that new markers for the early diagnosis of myocardial ischemia [41] or necrosis of cardiac cells [42] might be used for early detection of cardiac dysfunction [43].

Therefore, the aim of the present study was to examine the acute effect of an ultra-endurance performance on cardiac biomarkers and cardiovascular function, which would allow early detection of myocardial dysfunction during and after a 24 h ultra-marathon race. A secondary aim was to study the variation of cardiac biomarkers in key points (i.e., at 42 km and 12 h) of the race in order to make inferences about the potential impact of a marathon race and a 12 h ultra-marathon race on these markers, too. It was hypothesized that the H-FABP and IMA might be more sensitive markers than standard cardiac biomarkers in the early detection of exercise induced myocardial injury, especially in ultra-marathon races.

## 2. Experimental Section

### 2.1. Participants

Study members were recruited from all the competitors of the ultra-marathon held during the 24-h Running Polish Championships. Enrolling process was conducted one week before the start of the race. The inclusion criteria were participation in at least five marathons before indexed event and written informed consent to take part in the study. Before the start of the run, all the ultra-marathoners were assessed for body composition using a model In Body220 (Biospace Inc., Seoul, Korea) analyzer and completed a survey on their training history. During the run and immediately after finishing, the participants were under medical care. The study protocol was approved by the Ethics Committee of the Academy of Physical Education in Katowice, Poland, and conformed to the standards set by the Declaration of Helsinki.

The runners competed on a 3000-m oval running track located in a natural flat terrain. Weather conditions at the start of the race (12 p.m.) were good (20 °C and 53% relative humidity), but they changed during the race and reached 12 °C and 94% relative humidity at the finish of the race. The runners were constantly running with two 10-minute breaks to collect blood samples. During the race the runners were allowed free (unrecorded) intake of food and liquids and were also allowed to take short rest breaks at their free will. The total distance covered of each participant was recorded by an electronic chip, which allows automatic counting of the passages through the electronic gates set up on the track.

Two weeks before the race, all subjects performed a standard incremental treadmill exercise test (LE 200 treadmill, Jaeger, Frankfurt, Germany) to measure their individual aerobic performance (maximal oxygen uptake VO_2_max). The test started with a 3-min warm-up at 6 km/h and 0° inclination; the intensity was then increased by 2 km/h every 3 min up to maximal exercise intensity or volitional fatigue. Heart rate (HR) (PE-3000 Sport-Tester, Polar Inc., Kempele, Finland) and systolic and diastolic blood pressure (SBP/DBP) were measured (HEM-907 XL, Omron Corporation, Kyoto, Japan) before and immediately after the test. Pulmonary ventilation (VE), oxygen uptake (VO_2_), and carbon dioxide output (CO_2_) were measured continuously from the 6th minutes prior to exercise test and throughout each stage of the exercise test using the Oxycon Apparatus (CareFusion Germany 234 GMBH, Hoechberg Jaeger, Germany). Anthropometric and physiological characteristics of the participants are presented in Table 1.

### 2.2. Echocardiography

All subjects underwent baseline echocardiography assessment at least 10 days prior to the ultra-marathon and 48-h post-race. Parasternal and apical views were obtained using a standard echocardiography machine (M-mode and two-dimensional Doppler echocardiography; GE Vivid E9 General Electric, Oslo, Norway) with standard imaging transducers. All analyses were made and/or supervised by the same cardiologists experienced in the technique. Left ventricular end-diastolic dimensions (LVEDD) were measured at the onset of the QRS complex. The left ventricular (LV) and right ventricular (RV) volumes were derived according to the modified Simpson’s method. The left ventricular ejection fraction (LVEF) was calculated in the standard fashion from LV end-diastolic and end-systolic volume. Interventricular septal diastolic diameter (IVSDD), left ventricular posterior wall thickness dimension (LVPWTD), and left ventricle mass index (LVMI) indexed to body surface area were also determined [44]. Left ventricular mass was calculated using the area-length method according to guidelines of the American Society of Echocardiography [45]. In each subject relative wall thickness (RWT) was calculated as according to the rule: RWT = IVSDD + LVPWTD/LVEDD [46]. Pulsed Doppler imaging was used to determine LV inflow pattern. The mitral inflow E and A-wave velocities were measured and the E/A ratio was calculated [46,47].

### 2.3. Biochemical Analyses

Samples of venous blood were taken at five time points: i.e., 3 h before the race (Pre-race), three time points during the race: after completing the marathon distance (Marathon), after 12 h (12-h run), immediately after the 24-h ultra-marathon (24-h run) and 48 h after completion of the race (48-h post-race). Blood samples were managed according to the specifications of laboratory tests manufacturers in a core laboratory. Within 1 h of blood taking, the samples were carried to a nearby clinical laboratory for processing and analyses.

N-terminal pro-brain-type natriuretic peptide (NT-proBNP) levels were assayed using immunoassays method (SEA485Hu 96 Tests, Cloud-Clone Corporation, Katy, TX, USA). The measurements of cardiac troponin T (cTnT) were performed using immunoassays (SED22HU 96 Tests, Cloud-Clone Corporation, Houston, TX, USA). Serum activities of creatine kinase myocardial band (CK-MB) were measured using the immune-inhibition method (BioMaxima, Lublin, Poland). Serum high-sensitivity C-reactive protein (hs-CRP) was assessed using an immune-enzymatic method (Labor Diagnostika Nord GmbH & Co. KG, Nordhorn, Germany). Concentrations of ischemic modified albumin (IMA) were determined by immune-enzymatic method (CUSABIO BIOTECH CO., LTD, Wuhan, China). The heart fatty acid-binding protein (H-FABP) levels were determined using an immune-enzymatic method (Human H-FABP HK401 Elisa Kit, Hycult Biotech, Uden, The Netherlands).

### 2.4. Statistics

Shapiro-Wilk, Levene’s and Mauchly’s tests were used in order to verify the normality, homogeneity and sphericity of the sample’s data variances, respectively. Descriptive statistics were calculated and the results were presented as means and standard deviations or median and percentiles, according to the distribution of data. The data were analyzed by one-way repeated measures ANOVA. The significance of the differences between the variables was verified with the post-hoc test. Correlation coefficients between all the variables were determined with Pearson’s rank order test. The Statistics package v. 12 (StatSoft, Cracow, Poland, 12.0) was used for data processing and analyses. The differences were considered statistically significant at *p* < 0.05.

## 3. Results

From the total of 60 participants in the 24-h ultra-marathon, 14 subjects were included for analysis. The average distance covered during the race was 149.4 ± 33 km. Physical performance characteristics of the study participants are presented in Table 1.

The mean running velocities were associated with running distance and decreased during the race (Table 2). There were marked differences in individual race performance measures; however, running velocity was not significantly related to age, body mass and BMI. Eight participants finished the race with a distance of over 150 km (173.0 ± 22.9 km) and were enrolled in a long-distance group (Group 1). Runners with shorter distance (117.7 ± 9.6 km) have been included in the short-distance group (Group 2). The results of echocardiographic examinations of the heart are presented in Table 3. Resting heart rate (HR) and blood pressure did not differ between pre- and post-race. No significant post-race changes were observed in either LV or RV dimensions on completion of the ultra-marathon. The LV ejection fraction tends to be higher post-ultra-marathon compared to baseline values. Diastolic function was altered post-race with a significant reduction in E wave velocity (peak early diastolic filling), a significant increase in A wave velocity (peak late diastolic filling) resulting in significant decrease in E/A ratio (Table 3).

NT-proBNP levels increased in ultra-marathoners during the race, reached a maximum at the marathon distance (*p* < 0.01) and decreased at the end of the observational period (Figure 1). NT-proBNP levels correlated moderately with age (*p* < 0.01), 24-h run NT-proBNP levels correlated negatively with running distance (*p* < 0.01) (Table 4). There was a constant increase in mean concentration of CK-MB with a peak at the race finish with concentrations significantly higher than at baseline (*p* < 0.001), at post-marathon (*p* < 0.001) and at 12-h run (*p* < 0.001) (Figure 2). Concentration of CK-MB measured after the marathon distance correlated with RWT (*r* = 0.55, *p* < 0.03). There was a correlation between 24-h run CK-MB concentration and LVESD (*r* = 0.60, *p* < 0.03). Mean serum cTnT concentrations showed an increasing trend during the run, with a significant difference between 24-h run compared to concentration at baseline (*p* < 0.001), marathon (*p* < 0.001) and 12-h run (*p* < 0.01) (Figure 3). Marked running-related increases were found in serum mean levels of CRP during the first 12 h/24 h of the race, (*p* < 0.01 and *p* < 0.001, respectively). Significantly higher serum CRP levels were observed post 24-h run compared to Marathon (*p* < 0.001) and 12-h run (*p* < 0.001) (Figure 4). A significant correlation was found between 24-h run CRP levels and subject’ age (*p* < 0.02) and running distance (*p* < 0.05) (Table 4).

No significant changes were observed in response to serum IMA concentrations (Figure 5). Significant correlations were found between Marathon IMA levels and NTproBNP (*r* = 0.71, *p* < 0.01) and NTproBNP levels and LVMI (*r* = 0.57, *p* < 0.03). A negative correlation between IMA Marathon levels and running distance (*r* = −0.65, *p* < 0.01) was observed. The most marked changes in cardiac injury-related indices were the increase in circulating H-FABP levels during the 12-h/24-h run (*p* < 0.001) compared to pre-race values. Significantly higher H-FABP levels were observed during the 12-h run and 24-h run compared to marathon distance (*p* < 0.01 and *p* < 0.001, respectively) (Figure 6). Significant correlation was found between post ultra-marathon (24-h run) CKMB levels and H-FABP (*r* = 0.64, *p* < 0.05). As expected, H-FABP levels correlated with RWT and running distance (*r* = 0.56, *p* < 0.05, and *r* = 0.59, *p* < 0.05, respectively) (Table 4). No significant differences were observed between the long-distance runners (Group 1) and the short distance runners (Group 1) in CRP, CKMB, IMA, and cTnT concentrations recorded before and in response to 24-h run (*p* >0.05). Significant lower NT-proBNP levels were observed in response to 24-h run in runners with higher distance (104.0 ± 40.6 vs. 260.8 ± 147.6 pg/mL; *p* < 0.05, respectively) and at 48 h post-race (115.1 ± 39.7 vs. 162.5 pg/mL, *p* < 0.02, respectively) compared to runners with lower distance.

## 4. Discussion

In this study, an impact of an ultra-marathon on the range of standard (e.g., NT-proBNP, cTnT, CK-MB) and novel (IMA, H-FABP) cardiac biomarkers was analyzed which we hypothesized would allow for early detection of myocardial dysfunction during and after 24-h ultra-marathon races. The main findings in this study were that the markers of myocardium ischemia serum concentrations were significantly higher post 24-h run as compared to pre-race levels.

The current study confirmed a previous finding that running ultra-marathon was associated with a transient increase in cardiac biomarkers as a sign of exercise-induced myocardium overload [19,33,48]. However, the most interesting finding of our study was that the dynamic of secretion biomarkers linked to cardiac injury were differentially regulated during a 24 h ultra-marathon. This might confirm the greatest importance of the cardiac biomarkers release early to the bloodstream which are be useful for both rapid confirmation and exclusion of myocardium injury.

In our data, we have analyzed associations between exercise-induced changes in cardiovascular risk markers and echocardiography variables in response to a 24 h ultra-marathon. Our data support the hypothesis that ultra-marathon running leads to significant decreases in transmitral velocity (E/A ratio) indicating an inherent alteration in LV relaxation. The indices of LV and RV systolic and diastolic functions had normalized and returned to pre-race levels after 48 h post-race.

### 4.1. Echocardiography Examinations

In our study, the echocardiographic examinations of the left ventricular structure (e.g., left ventricular mass and left ventricular mass index) showed significantly higher values compared to normal heart dimensions, clearly indicating adaptive heart hypertrophy [49]. As expected, left and right ventricular systolic and diastolic dimensions showed no significant differences after completion of the ultra-marathon compared to pre-race values. Diastolic function was altered post-race characterized by a reduction in peak early diastolic filling, an increase in peak late diastolic filling and a resultant decrease in E/A ratio post-race compared to baseline values. LV diastolic functionality post-race changes were not significantly associated with the decreased post-race LVEF in contrast to previous studies [50].

A possible explanation of this feature relates to the volume overload of LV, which is greater with higher exercise intensity and VO_2_max [33]. In elite athletes with LV hypertrophy, it has been previously shown that detraining is associated with a reduction in early diastolic filling and a consequent reduction in the E/A ratio [14]. It has been evidenced that the results of Doppler examination of mitral inflow might be affected by several factors including blood pressure [13], volume status of LV and RV [3,14], dehydration [19] and redistribution of blood flow [4]. Therefore, the possible mechanism to this feature relates to the preload, which may have contributed to further elongation of myocardial fibers and LV diastolic function. Nevertheless, there could be other regulatory mechanisms, which in combination may have decreased LV diastolic function and can be interpreted as “cardiac fatigue”.

### 4.2. Cardiac Biomarkers

This study revealed that running a marathon had a significant impact on NTproBNP levels, whereas competing in a 24 h ultra-marathon significantly increased hypoxia–induced biomarkers (i.e., cTnT, CKMB, HFABP) compared to baseline levels. It was previously documented that prolonged strenuous exercise may be associated with adverse cardiovascular consequences and higher release of cardiac biomarkers.

In our study, the baseline serum NT-proBNP was within a normal range in all runners, however, it showed a marked, over more than two-fold elevation during the marathon. NT-proBNP is an indicator of the physiological consequences of hypoxia and illustrates the activation of the neuroendocrine system and early hemodynamic disturbances. An increase in concentration may occur in cases of elevated pressure filling and provide a reliable diagnostic and prognostic information [51]. Contrary to our expectations, we found no significant differences in the expression of NT-proBNP during long-term run and post-24 h race (Figure 1). Notably, during the race individual serum NTproBNP levels exceeded the limits accepted pre-race for the detection of heart failure (125 pg/mL). NT-proBNP post-race levels correlated positively with age but inverse correlations were found in response to the 24-h running distance. Significant lower NTproBNP levels were observed in response to 24-h run and 48 h post-race levels in runners with higher running distance compared to runners with lower distance. These results suggest that the greater performance level may have a potential beneficial effect of cardiac function improvement in ultra marathon runners.

Similar changes have been documented in recreational athletes completing a marathon. It has been noted that post-marathon values of NT-proBNP correlated with alterations in left ventricle diastolic function [16,21,26]. High NT-proBNP levels were also observed in response to 100 km, 200 km and 308 km ultra-marathon running compared to pre-race levels [52]. It has been suggested that an increase in NT-proBNP levels probably resulted from continuous hemodynamic overload and/or impaired LV relaxation might represent a physiological myocardium protective response [51]. Notably, in our study, post 24-h run values of NT-proBNP were independently associated with running distance suggesting that the level of preparation undertaken by these ultra-marathoners might have stabilized cardiac function as well as NT-proBNP levels in response to long-term running.

Several studies investigated conventional biomarkers such cTnT and CK-MB release primarily associated with hypoxia–induced myocardial damage [12,36,53]. Recent data also included novel biomarkers primarily associated with pressure overload and stretch of myocardial fibers [41,54,55]. The role of these biomarkers has been extensively studied with interpretation for consequences of leakage from cellular membranes of myocyte necrosis [56].

The results of this study are in line with the common knowledge that prolonged endurance exercise causes increases in cardiac markers of hypoxia and muscle damage [8,17,19]. In our study group, CK-MB activity and cTnT levels increased significantly only in response to long-term running (24-h run) (Figure 2 and Figure 3). Contrary to our expectations, we found no significant differences in the expression of these cardiac injury-related markers in ultra-marathoners after the marathon distance compared to pre-race or in after 12 h of running. The increases in CKMB in this study were negative correlated to echocardiography parameters of left ventricular systolic function.

Interestingly, no significant effects were found in the ischemia-modified albumin (IMA) levels after 24 h of running (Figure 5) with increasing trend to higher values during marathon and 12-h run. The lack of significant higher IMA levels after ultra marathon distance vs. pre-race vales could be explained by the presence of other mechanisms responsible for training-induced cardio protection include increased shock proteins levels or increased myocardial antioxidant capacity in elite endurance trained athletes. Previous studies reported IMA is an early marker of myocardial ischemia in the diagnosis of coronary artery disease [28,41]. It was revealed that IMA levels may increase during myocardial ischemia as seen during strenuous exercise [27]. A release of this marker may result after an increased oxidative stress caused by ischemia reperfusion injury or other mechanisms responsible for a reduction in the coronary blood flow [54]. It is worth to note that previous study also demonstrated that IMA levels decrease after physical exercise, and it was hypothesized that this decrease may have been attributed to hemoconcentration or lactate levels [54]. Further studies are required to identifying the role of this cardiac biomarker for the diagnosis of myocardial ischemia following ultra-endurance exercise.

It seems that LV dysfunction and long-term endurance exercise may imply a myocardial injury and were both responsible for an elevated release of the cardiac biomarkers [10,31]. Importantly, echocardiographic and cardiac biomarkers analyses in this study group, obtained after 48 h of completion of the ultra-marathon demonstrated that most of these indices of cardiac function had normalized.

It is generally accepted that CKMB increases result from enzyme leakage from cardiac muscles due to mechanical stretch or increased membrane permeability [21,24]. The results of the presented study support the suggestion that long-term exercise-related release of these markers does not reflect a clinically threatening heart injury, but are due to transient and reversible increases in myocardial sarcolemma permeability with no myocardial damage [26,56].

An increase in the concentration of markers of cardiac injury in response to acute extreme physical exercise has been observed previously; however, it has been shown that well trained athletes were less likely to experience this increase compared to sedentary obese untrained women [57]. Due to extreme continuous exercise program, ultra-marathon runners presented increased myocardial mass, a typical characteristic of “athlete’s heart”. Accordingly, routine testing for cardiac biomarkers in this group might occasionally present false-positive outcomes towards myocardial ischemia, whereas lifestyle interventions such as exercise training did not result in increase in cardiac markers [58]. It has been debated that continuous exercise program for patients with increased level of hs-troponin at rest (or mild physical activity) in hypertrophic cardiomyopathy might induce myocardial damage and lead to adverse events in longer perspective [59,60]. Considering highly trained ultra-marathon runners, the levels of cardiac markers should be interpreted with regard to increased LVMI and hypertrophic remodeling characteristic for athletes but should not be crosslinked with hypertrophic cardiomyopathy which represented certain pathology [61]. It still remained to be determined whether this might be extrapolated to indications for systematic moderate physical activity among general population and whether any screening towards LVH or increased baseline cardiac markers would be needed. 

On the other hand, it should be highlighted that exercise training might have negative effect in hypertrophic cardiomyopathy (HCM) patients. HCM was differentiated from “athlete’s heart” presenting (i) a LV end-diastolic diameter larger than the normal when the LV ejection fraction was 50% and (ii) an early impairment of LV diastolic function [62]. Moreover, HCM patients showed larger carotid and brachial arterial wall thickness, and slower brachial artery peak blood flow response following forearm ischaemia [63]. It has been suggested that exercise might be contraindicated in HCM patients with painless and painful ischemia (i.e., positive hs-troponin at rest), where such a problem might be present in 50% of HCM patients and might be related with same abnormal echocardiography and Holter findings [64]. A decreased level of exercise capacity in these patients might be due to increased BMI [65]. Thus, exercise training should be carefully prescribed (e.g. gradual increase of exercise intensity during training period) to HCM patients in order to achieve functional improvement [66].

Consistent with earlier reports [5,8], there was a progressive rise in blood hsCRP that is involved in the induction of anti-inflammatory cytokines in circulating monocytes and is responsible for the recognition and removal of damaged cells [67]. In our study, plasma hsCRP concentration during the first marathon distance was fairly stable. Changes in plasma hsCRP content occurred later, i.e., after finishing 12 h when it rose in an exponential pattern to reach at the finish of a 24-h race a level more than 15 times higher than the pre-race value (Figure 4). The significant correlation between plasma CRP and running distance in our study supports the presumption of the role of inflammatory process as a primary inducer of hepatic production of acute phase proteins during long-term endurance exercise [68].

One important observation made is this study was that in ultra-marathoners the heart fatty acid-binding protein (H-FABP) has been found to have different release times and lower times of reaching maximal concentrations during the 24-h ultra-marathon compared to other markers of myocardial injury. Recent investigations indicated that H-FABP is a sensitive and early marker of myocardial damage [69,70]. It was reported that H-FABP levels started increasing 1 h following myocardial cell damage with peak levels between the 6th and the 8th hour and may be more effective for detecting post-exercise marker of cardiac injury [71]. In this study, significantly higher H-FABP levels were observed post 12-h run and 24-h run compared to post-marathon distance. As expected, significant correlation was found between post 24-h run H-FABP levels and CKMB activity as well as running distance. It has been suggested that the much lower molecular weight (H-FABP is 15 kDa) than CKMB and cTnT (86 kDa vs. 33 kDa, respectively) might especially cause H-FABP higher molecules membrane permeability increase. As a result, this condition may lead to an enhancement in the plasma level of this biomarker at an early phase of the exercise. The associations found between actual running distance and serum H-FABP levels may imply that duration were responsible for elevated release of the cardiac biomarkers.

It could be assumed that long-term endurance exercise may stabilize the secretion of selected biomarkers associated with myocardial pressure overload, potentially leading to improved cardiovascular function [62]. By contrast, high levels of biomarkers for early diagnosis of cardiac ischemia may contribute to the risk of hypoxia-induced myocardial injury in ultra-marathoners. However, further research may be warranted to explore and find effective methods for preventing chronic cardiovascular complications associated with long-term running. Although our results are consistent with recent reports, our study was limited by a small sample size; however, the research design is dictated by the subject’s exercise tolerance.

## 5. Conclusions

In summary, the 24-h ultra-marathon-related strain caused an acute but transient cardiac dysfunction. These phenomena were evidenced by high levels of cardiac biomarkers as a sign of exercise-induced myocardium overload, but not associated with echocardiography parameters of myocardial dysfunction. Interestingly, the dynamic of secretion biomarkers linked to cardiac injury were differentially regulated during the 24-h ultra-marathon. It might be suggested that both exercise duration and intensity may play a crucial role in cardiovascular adaptive mechanisms and cause a higher risk of cardiac stress in ultra-marathoners. However, it should be emphasized that the apparent lack of evidently unfavorable, permanent effects of the ultra-endurance effort in our study may not reflect the actual risk from such activity in general population. This is because the ultra-marathon runners represented a particularly fit subgroup of those who systematically attempt this type of physical activity.

## Figures and Tables

**Figure 1 jcm-08-00057-f001:**
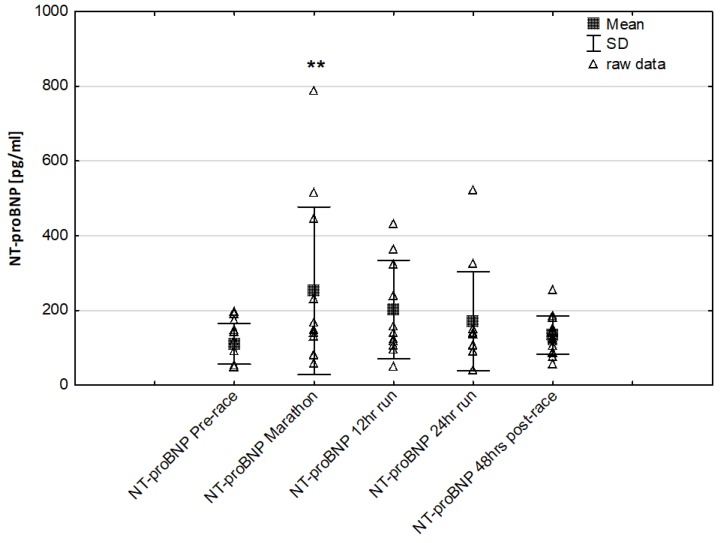
Ultra-marathon-induced changes in serum N-terminal pro-brain natriuretic peptide (NT-proBNP) ** *p* < 0.01 versus the corresponding pre-race value (Pre-race).

**Figure 2 jcm-08-00057-f002:**
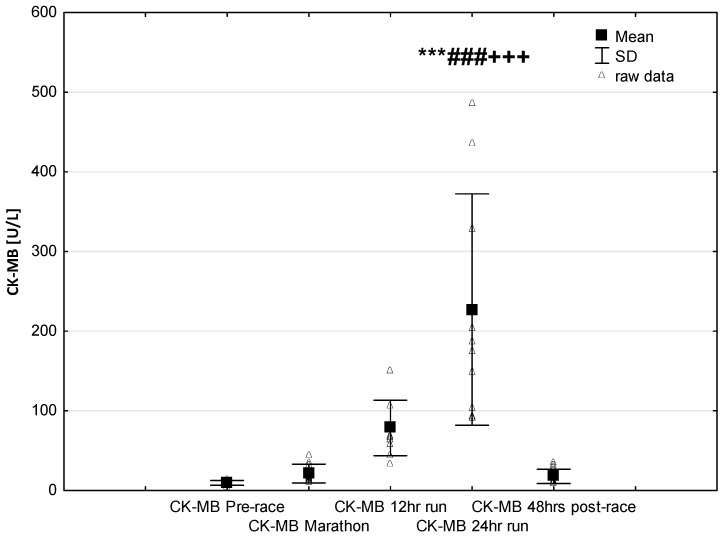
Ultra-marathon-induced changes in serum creatinine kinase-myocardial band (CK-MB). *** *p* < 0.001 versus the corresponding pre-run value (Pre-race); ### *p* < 0.001 versus the corresponding post-marathon value (Marathon); +++ *p* < 0.001 versus the corresponding post 12-h run value (12-h run).

**Figure 3 jcm-08-00057-f003:**
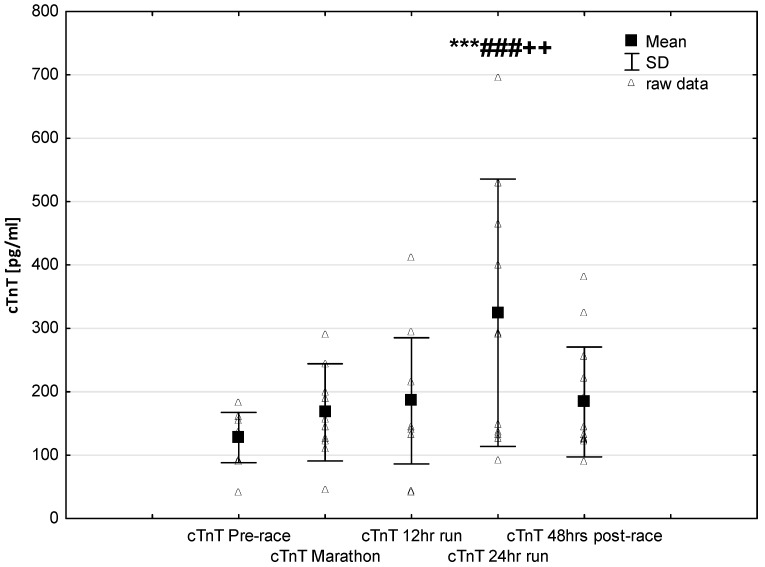
Ultra-marathon-induced changes in serum cardiac specific troponin T (cTnT). *** *p* < 0.001 versus the corresponding pre-race value (Pre-race); ### *p* < 0.001 versus the corresponding post-marathon value (Marathon); ++ *p* < 0.01 versus the corresponding post 12-h run value (12-h run).

**Figure 4 jcm-08-00057-f004:**
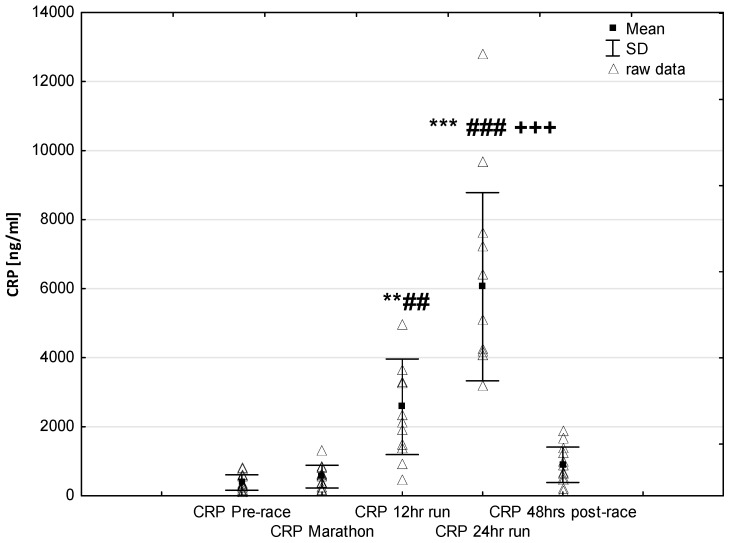
Ultra-marathon-induced changes in serum high sensitive C-reactive protein (hsCRP). ** *p* < 0.01, *** *p* < 0.001 versus the corresponding pre-race value (Pre-race); ## *p* < 0.01, ### *p* < 0.001 versus the corresponding post-marathon value (Marathon); +++ *p* < 0.001 versus the corresponding post 12-h run value (12-h run).

**Figure 5 jcm-08-00057-f005:**
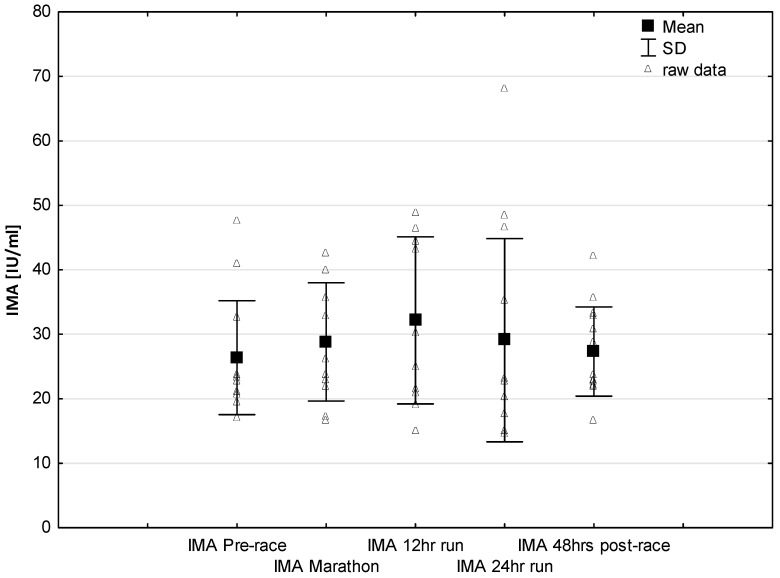
Ultra-marathon-induced changes in serum ischemia modified albumin (IMA).

**Figure 6 jcm-08-00057-f006:**
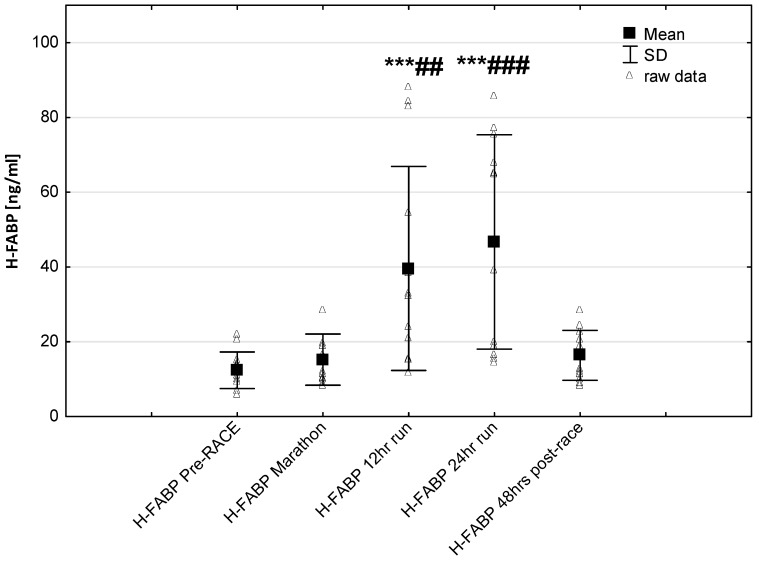
Ultra-marathon-induced changes in serum heart-type fatty acid binding protein (H-FABP). *** *p* < 0.001 versus the corresponding pre-race value (Pre-race); ## *p* < 0.01, ### *p* < 0.001 versus the corresponding post-marathon value (24-h run).

**Table 1 jcm-08-00057-t001:** Physical performance characteristics of the study participants.

Variables	Participants, *n* = 14
Age (years)	40.0 ± 11.7
Body mass (kg)	78.4 ± 11.4
Body height (cm)	178.0 ± 5.0
BSA (m^2^)	2.0 ± 0.2
BMI (kg·m^−2^)	24.5 ± 2.4
VO_2_max (mL·min^−1^·kg^−1^)	55.3 ± 8.8

BSA: body surface area, BMI: body mass index, VO_2_max: maximal oxygen uptake.

**Table 2 jcm-08-00057-t002:** Ultra-marathon race performance measures.

Variables	Mean	Min–Max
Marathon time (h)	4.8 ± 0.7	4.4–5.4
Marathon running velocity (km/h)	8.6 ± 0.8	7.8–9.5
12 h distance (km)	83.4 ± 6.9	57.4–120.4
12-h running velocity (km/h)	7.0 ± 0.6	4.8–9.9
24 h distance (km)	149.4 ± 33.0	100.7–214.5
24-h running velocity (km/h)	6.2 ± 1.4	4.2–8.9

**Table 3 jcm-08-00057-t003:** Echocardiographic variables of the subjects (mean ± SD).

Variables	Pre Ultra-Marathon (Pre-Race) *n* = 14	Post-Ultra-Marathon (48 h Post-Race) *n* = 14
LVM (g)	271.6 ± 37.7	271.8 ± 38.2
LVMI (g/m^2^)	138.0 ± 11.6	138.0 ± 11.2
LVEDD (mm)	51.5 ± 0.4	51.3 ± 0.3
LVESD (mm)	29.6 ± 0.4	29.4 ± 0.6
IVSDD (mm)	12.3 ± 0.7	12.3 ± 0.6
LVPWTD (mm)	10.5 ± 0.7	10.6 ± 0.7
RWT	0.44 ± 0.04	0.45 ± 0.05
LA (mm)	39.0 ± 8.5	38.0 ± 10.3
RVDD (mm)	29.0 ± 3.0	29.0 ± 3.2
LVEF %	61.4 ± 5.5	64.6 ± 8.5
E/A	1.9 ± 0.3	1.3 ± 0.1 **
SBP (mm Hg)	124.4 ± 14.5	120.0 ± 10.5
DBP (mm Hg)	77.0 ± 11.0	77.0 ± 9.0
HR (beats/min)	58.0 ± 10.0	62.0 ± 8.0

BMI—body mass index, LVM—left ventricular mass, LVMI—left ventricular mass index, LVEDD—left ventricular end-diastolic dimension, LVESD—left ventricular end-systolic dimension, IVSDD—intraventricular septum diameter during diastole, LVPWTD—left ventricular posterior wall thickness during diastole, RWT—relative wall thickness, LA—left atrial anterior-posterior dimensions, RVDD—right ventricular diastolic dimensions LVEF—left ventricular ejection fraction, E/A—ratio of transmitral E to transmitral A, SBP—systolic blood pressure; DBP—diastolic blood pressure, and HR—heart rate. ** *p* < 0.01 versus the pre-ultra-marathon value.

**Table 4 jcm-08-00057-t004:** Correlations between cardiac biomarkers age and running distance measures.

Variables (24-h run)	Age	Running Distance
NT-proBNP (pg/mL)	*r* = 0.66 *p* < 0.01	*r* = −0.69 *p* < 0.01
CK-MB (U/L)	*r* = 0.47 *p* < 0.87	*r* = 0.39 *p* < 0.17
IMA (IU/mL)	*r* = 0.14 *p* < 0.62	*r* = −0.42 *p* < 0.14
cTnT (pg/mL)	*r* = −0.77 *p* < 0.79	*r* = −0.72 *p* < 0.49
CRP (ngmL)	*r* = −0.61 *p* < 0.02	*r* = 0.52 *p* < 0.05
H-FABP (ng/mL)	*r* = −0.19 *p* < 0.52	*r* = 0.59 *p* < 0.05

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
