# Peer review of "Acute Responses of Novel Cardiac Biomarkers to a 24-h Ultra-Marathon"

_jcm, 2019, doi:10.3390/jcm8010057_

Reviewer 1 Report

The manuscript by Żebrowska et al. examines the acute effect of an ultra-endurance performance and its impact on 6 different cardiac markers and cardiac function at various timepoints, pre-, during- and post- run.

1) The naming of the time-points used in the study is not always clear and misleading. According to the methods: Samples of venous blood were taken at five time points: i.e., 3 hrs. before the race, three time points during the race: after completing the marathon distance, after 12 hrs., immediately after the 24-hour ultra-marathon and 48 hrs. after completion of the run. In the results section and the graphs, the names of the samples are: Pre-Run, Post-Marathon, Post 12h Run, Post 24h Run, Post-Run 48h.

The description of the time points in the methods section is clear but their naming in the results/graphs is slightly misleading. For example, while the 12hr sample is taken during the race (ie 12hrs of running) name ‘post 12h Run’ used in the results/graphs can be mistaken for 12 hours after the race. The same applies for the 24hr sample. Maybe they should be renamed to ‘12hr run’ and ‘24hr run’ samples. Similarly, the post-Marathon could be named ‘Marathon’ or ‘42km’ (as mentioned at page 3 line 87). In this way, it will be clear to the reader that these samples are taken during the race and they are very different timepoints to the post-48 run sample, which is taken 48 hrs after the race.  

 2) It would be very helpful if any correlations with clinical parameters described in the text of the results (eg for NT-proBNP levels and age, running distance or CK-MB with cardiac function parameters) are provided in table-format.

3) In Table 2, the order of the Marathon running velocity min-max values should be 7.8-9.5 instead of 9.5-7.8.

4) There is considerable variation in the Min-Max race performance values presented in Table 2 with regards to time, distance and velocity of the runners, especially at 12hr and 24hr timepoints. Do the authors feel that it is reliable to include subjects with such variation and compare their biomarkers? For example, in the 12hr running group how can one include in the same group a subject that has run 57.4km and one that has run 120.4km and expect to find similar changes in their biomarkers?

Along these lines, there is actually a recent paper describing ‘Muscle damage and inflammation biomarkers after two ultra-endurance mountain races of different distances: 54 km vs 111 km’ reporting differences in markers associated with inflammation, muscle injury and cardiac damage among these two groups.

It may be better to subgroup athletes with similar time/distance/velocity within a timepoint so as to have a more homogeneous population of samples. Also, information from athletes with Min- and Max- performance values could be used to extrapolate findings regarding the impact of duration and intensity on cardiac stress in these ultra-marathoners.

5) Given that there is such variation in performance values, which can most likely explain for some of the considerable variation observed in biomarkers measurements within a single timepoint, the authors should consider using scatter plots instead of graphs to present their findings.

6) How do the authors explain the lack of change in IMA levels?

7) Given that the main findings in this study were that the markers of myocardium ischemia serum concentrations were significantly higher post 24h race (ie at the end of the race) as compared to pre-race levels, do the authors believe that they have achieved their aim of identifying cardiac biomarkers for early detection of myocardial dysfunction during and after 24-h ultra-marathon?

8) The point raised by the authors in the last sentence of the conclusion regarding the fitness of the ultra-marathon runners who are systematically in this type of physical activity is really important and should also be mentioned in the Discussion. Could the fact that the main changes in biomarkers were observed at the 24hr sample be due to the extreme physical condition of these athletes?    

Minor:

In lines 260 and 264, please correct misspelling of ‘24-hourh’

Author Response

Response to Reviewer 1:

 1) The naming of the time-points used in the study is not always clear and misleading. According to the methods: Samples of venous blood were taken at five time points: i.e., 3 hrs. before the race, three time points during the race: after completing the marathon distance, after 12 hrs., immediately after the 24-hour ultra-marathon and 48 hrs. after completion of the run. In the results section and the graphs, the names of the samples are: Pre-Run, Post-Marathon, Post 12h Run, Post 24h Run, Post-Run 48h.

 The description of the time points in the methods section is clear but their naming in the results/graphs is slightly misleading. For example, while the 12hr sample is taken during the race (ie 12hrs of running) name ‘post 12h Run’ used in the results/graphs can be mistaken for 12 hours after the race. The same applies for the 24hr sample. Maybe they should be renamed to ‘12hr run’ and ‘24hr run’ samples. Similarly, the post-Marathon could be named ‘Marathon’ or ‘42km’ (as mentioned at page 3 line 87). In this way, it  will be clear to the reader that these samples are taken during the race and they are very different time points to the post-48 run sample, which is taken 48 hrs after the race.

Answer: We appreciate this comment that the naming of the time-points used in the study to analyze the blood samples is not clear and is slightly misleading. The description of the time points was consequently changed in the manuscript according suggestion to be clear to the reader: Samples of venous blood were taken at five time points: i.e., 3 hrs. before the race (Pre-race), after completing the marathon distance (Marathon), after 12 hrs. run (12 hr run), immediately after the 24-hour ultra marathon (24 hr run), and 48 hrs. after the race (48hrs. post-race).

 2) It would be very helpful if any correlations with clinical parameters described in the text of the results (eg for NT-proBNP levels and age, running distance or CK-MB with cardiac function parameters) are provided in table-format.

Answer: We agree with the expert reviewer and the most important correlations were provided in table 4. The description of the results has been corrected in the results section.

 3) In Table 2, the order of the marathon running velocity min-max values should be 7.8-9.5 instead of 9.5-7.8.

Answer: We agree with the expert reviewer and the Table 2 has been corrected according to suggestion.

 4) There is considerable variation in the Min-Max race performance values presented in Table 2 with regards to time, distance and velocity of the runners, especially at 12hr and 24hr time points. Do the authors feel that it is reliable to include subjects with such variation and compare their biomarkers? For example, in the 12hr running group how can one include in the same group a subject that has run 57.4km and one that has run 120.4km and expect to find similar changes in their biomarkers? Along these lines, there is actually a recent paper describing ‘Muscle damage and inflammation biomarkers after two ultra-endurance mountain races of different distances: 54 km vs 111 km’ reporting differences in markers associated with inflammation, muscle injury and cardiac damage among these two groups. It may be better to subgroup athletes with similar time/distance/velocity within a time point so as to have a more homogeneous population of samples. Also, information from athletes with Min- and Max- performance values could be used to extrapolate findings regarding the impact of duration and intensity on cardiac stress in these ultra-marathoners.

Answer: We appreciate this comment that the variation in the min-max race performance values might be associated with different changes in cardiac biomarkers levels. Therefore, a more a relevant method to assessing the cardiac markers levels is to compare the effects of 24 hrs. running in two groups. However, the values of the CRP, CKMB, IMA, cTnT recorded before and in response to 24 hr run did not differentiate the participants with high distance ( Gr 1 n=8 distance > 150 km) and in response to low distance (Gr 2 n= 6 < 150 km). Analysis showed no significant difference between Gr 1 vs. Gr 2. The significant difference was only observed at NTproBNP concentrations. Significant lower NTproBNP levels were observed in response to 24 hr run (p<0.05) and 48 hrs. post race levels (p<0.02) in runners with higher distance compared to runners with lower distance. We have added the description and interpretation of these results to the manuscript. We explained the effects of performance level as a potential beneficial effect of cardiac function improvement. We wish to thank the Reviewer for the valuable suggestions and references.  

 5) Given that there is such variation in performance values, which can most likely explain for some of the considerable variation observed in biomarkers measurements within a single time point, the authors should consider using scatter plots instead of graphs to present their findings.

Answer: We agree with the expert reviewer and we used scatter plots instead of graphs to present the results. However, it is possible the graphical presentation of data is more readable to explain significant differences during ultramarathon race.

 6) How do the authors explain the lack of change in IMA levels?

Answer: We agree with the expert reviewer. One of the main findings is that marathon as well as 12 hr run and 24 hr run increased cardiac biomarkers levels, however to different extent. In our study mean serum IMA levels showed an increasing trend during marathon and 12 hr run. The lack of significant higher IMA levels after ultra-marathon distance vs. pre-race vales could be explained by the presence of other mechanisms responsible for training-induced cardio protection include increased shock proteins levels or increased myocardial antioxidant capacity in elite endurance trained athletes. The impact of these mechanisms on the diagnostic sensitivity of IMA will be able to be explained in further studies.

 7) Given that the main findings in this study were that the markers of myocardium ischemia serum concentrations were significantly higher post 24h race (i.e. at the end of the race) as compared to pre-race levels, do the authors believe that they have achieved their aim of identifying cardiac biomarkers for early detection of myocardial dysfunction during and after 24-h ultra-marathon?

Answer: We agree with the expert reviewer and clinical trials presented in our study of ultra-endurance exercise interventions in athletes demonstrated that these biomarkers would allow for early detection of myocardial dysfunction during and after 24-h ultra-marathon.

 8) The point raised by the authors in the last sentence of the conclusion regarding the fitness of the ultra-marathon runners who are systematically in this type of physical activity is really important and should also be mentioned in the Discussion. Could the fact that the main changes in biomarkers were observed at the 24hr sample be due to the extreme physical condition of these athletes?    

Answer: We agree with the expert reviewer and we added the sentences regarding the fitness of the ultra-marathon runners who are systematically in this type of physical activity is really important in the Discussion section.

 Minor:

In lines 260 and 264, please correct misspelling of ‘24-hourh’

Answer: We agree with the expert reviewer and the misspelling was corrected.

Reviewer 2 Report

The paper about acute responses of novel cardiac biomarkers to a 24-hour ultra-marathon is well planned and conceived. There are only some inaccuracies:

Introduction: lines 75-78, the concept reported in the sentence "Several cardiac… time-to-peak concentrations." is the same of the subsequent sentence in lines 78-81 "In previous studies… maximal concentrations". So it seems to me that the second one could replace the first sentence.

Experimental section: it is well conceived and exhaustive.

Results: line 183: the values relative to E/A ratio are reported in Table 3, so I think that it is not necessary to report them here too.

line 199: LVSV, what does it mean? I think it is Left Ventricular Stroke Volume, but this acronym has never been explained anywhere.

line 199: it seems to me more correct to write "Mean serum cTnT concentrations showed an increasing trend…" removing the word "significant", that correctly appears in line 200.

Line 236: the significance between brackets must be only p<0.001.< span="">

Table 3: in the legend there are no explanations of the abbreviations LVEDD and LVESD. Line 214: RWT instead of RVT.

Discussion: it is well written and exhaustive.

Author Response

Response to Reviewer 2:

 Comments and Suggestions for Authors

 The paper about acute responses of novel cardiac biomarkers to a 24-hour ultra-marathon is well planned and conceived. There are only some inaccuracies:

 Introduction: lines 75-78, the concept reported in the sentence "Several cardiac… time-to-peak concentrations." is the same of the subsequent sentence in lines 78-81 "In previous studies… maximal concentrations". So it seems to me that the second one could replace the first sentence.

Answer: We agree with the reviewer’s comment. The sentences have been corrected according to the suggestions.

 Experimental section: it is well conceived and exhaustive.

 Results: line 183: the values relative to E/A ratio are reported in Table 3, so I think that it is not necessary to report them here too.

Answer: We agree with the expert reviewer and we have changed the text

 line 199: LVSV, what does it mean? I think it is Left Ventricular Stroke Volume, but this acronym has never been explained anywhere.

Answer: We agree with the expert reviewer and the sentence has been changed and the LVSV value has been removed.

 Line 199: it seems to me more correct to write "Mean serum cTnT concentrations showed an increasing trend…" removing the word "significant", that correctly appears in line 200.

Answer: We agree with the expert reviewer and we have changed the text to be more precise in our statements

 Line 236: the significance between brackets must be only p<0.001.< span="">

Answer: We agree with the expert reviewer and the statistical significance was corrected.

 Table 3: in the legend there are no explanations of the abbreviations LVEDD and LVESD. Line 214: RWT instead of RVT.

Answer: We agree with the expert reviewer and the explanation of abbreviations has been corrected according to the suggestions.

 Discussion: it is well written and exhaustive.

Answer: We greatly appreciate the reviewer’s comments and suggestions.  No further changes are required.

Round  2

Reviewer 1 Report

Thank you for your response to my comments, I appreciate your willingness and effort in addressing them. I just had one minor comment on the revised manuscript. Could you please add the statistical significance symbols within the revised figures (Figures 1, 2, 3, 4 and 6), as at the moment they are only present in the figure legend. I believe that this would be really helpful to the readers. 

Author Response

We agree with the expert reviewer and changed the figures as requested